# Belgian *Anopheles plumbeus* Mosquitoes Are Competent for Japanese Encephalitis Virus and Readily Feed on Pigs, Suggesting a High Vectorial Capacity

**DOI:** 10.3390/microorganisms11061386

**Published:** 2023-05-25

**Authors:** Claudia Van den Eynde, Charlotte Sohier, Severine Matthijs, Nick De Regge

**Affiliations:** 1Exotic and Vector-Borne Diseases, Sciensano, Groeselenberg 99, 1180 Brussels, Belgium; charlotte.sohier@sciensano.be (C.S.); nick.deregge@sciensano.be (N.D.R.); 2Viral Re-Emerging Enzootic and Bee Diseases, Sciensano, Groeselenberg 99, 1180 Brussels, Belgium; severine.matthijs@sciensano.be

**Keywords:** *Anopheles plumbeus*, field-collected mosquitoes, vector competence, vectorial capacity, Japanese encephalitis virus

## Abstract

*Anopheles plumbeus*, a day-active mosquito known to feed aggressively on humans, was reported as a nuisance species near an abandoned pigsty in Belgium. Since Japanese encephalitis virus (JEV) is an emerging zoonotic flavivirus which uses pigs as amplification hosts, we investigated (1) whether *An. plumbeus* would feed on pigs and (2) its vector competence for JEV, to investigate whether this species could be a potential vector. Three- to seven-day-old F0-generation adult mosquitoes, emerged from field-collected larvae, were fed on a JEV genotype 3 Nakayama strain spiked blood meal. Blood-fed mosquitoes were subsequently incubated for 14 days at two temperature conditions: a constant 25 °C and a 25/15 °C day/night temperature gradient. Our results show that *An. plumbeus* is a competent vector for JEV at the 25 °C condition and this with an infection rate of 34.1%, a dissemination rate of 67.7% and a transmission rate of 14.3%. The vector competence showed to be influenced by temperature, with a significantly lower dissemination rate (16.7%) and no transmission when implementing the temperature gradient. Moreover, we demonstrated that *An. plumbeus* readily feeds on pigs when the opportunity occurs. Therefore, our results suggest that Belgian *An. plumbeus* mosquitoes may play an important role in the transmission of JEV upon an introduction into our region if temperatures increase with climate change.

## 1. Introduction

*Anopheles plumbeus* (Stephens 1828) mosquitoes are widespread in most European countries [1,2]. It is a day-active species known to persistently bite humans, with peak-activity at dawn and dusk.

*Anopheles plumbeus* is assumed to have contributed to malaria transmission in the past [1,3,4,5,6] and is a proven potential vector for West Nile virus, as transmission in a laboratory experiment was reported in the 1960s [7]. Because of its ecology as a woodland species, it has been considered of secondary importance as a vector species [1]. *Anopheles plumbeus* usually breeds in tree holes but is increasingly being associated with artificial breeding sites such as catch basins and septic tanks [1,2,3,4,6,8]. This leads to an expansion into urban and rural habitats which could increase its contact with domestic animals, livestock and humans. In some cases, their adaptation to artificial breeding sites leads to mass abundance and nuisance for humans in the surrounding area [9]. In that regard, it has presented serious nuisances in the past in the vicinity of an abandoned pigsty in Torhout, Belgium [4]. Next to humans, *An. plumbeus* also feeds on horses, cattle and birds [6,10,11] making it a potential bridge vector for arboviral transmission [10] and a relevant species to examine for their possible role in the transmission of various arboviruses.

Japanese encephalitis virus (JEV) is a zoonotic flavivirus that is maintained in a transmission cycle between mosquito vectors and vertebrate hosts, mainly Ardeid birds (herons and egrets) and pigs [12]. These hosts are considered replication hosts, meaning they produce high viremias after infection with JEV [13] allowing mosquitoes to potentially become infected when taking a blood meal. Moreover, pigs are known to have a high and prolonged viremia and are therefore considered the main amplification hosts of JEV [14]. Humans, cattle and horses are considered dead-end hosts, as JEV infection results in insufficient viremia to infect naive mosquitoes when taking a blood meal [12]. Nevertheless, infection of these hosts can lead to encephalitis in combination with fever, tremors, convulsions, coma and death [15]. In humans and mostly in children [16], 1% of infected individuals will develop encephalitis, with a mortality rate of 20 to 30% [17]. JEV is the principal cause of viral encephalitis in many countries in Asia with 68,000 cases reported annually [18] and is currently endemic in Southeast and East Asia, including the temperate zone of northeastern China, Japan, Korea [12,19] and the Torres Strait islands from where it spread to new regions in Australia in 2022 [20]. JEV is considered an emerging infectious disease. Urbanization, transportation, migration and movement of birds, climate change, wind currents and changes in land use have been favorable to its emergence and re-emergence [21]. For example, wind dispersal of infected mosquitoes and movement of viremic migratory birds most likely caused the spread of JEV in Australia in 2022 [20,22]. Additionally, transportation of infected mosquitoes might be accountable for the detection of JEV in an autochthonous human case in Angola and in a *Culex pipiens* mosquito pool in Italy, although the latter has never been formally confirmed [23,24] Thus, one possible scenario for the establishment upon introduction is that introduced infected mosquitoes lead to infection of susceptible animals. Another possibility is that viremic pigs or Ardeid birds are introduced. Native mosquitoes can become infected by taking a blood meal from these animals and, if competent, transmit JEV to the next host after an incubation period [25].

Whether a certain mosquito will be able to transmit JEV will depend on its vector competence. Vector competence is a parameter to be evaluated in the laboratory; it usually involves determining infection, dissemination and transmission [26]. For a species to be considered competent by these studies, it is necessary to establish transmission, i.e., presence of virus in the mosquitoes’ saliva. Vector competence is only one of the factors that determines whether a specific mosquito species will play a role in transmission under field conditions. Therefore, vectorial capacity must be taken into account; this includes additional factors such as vector density (abundance) relative to susceptible hosts, the probability that the vector will feed on a host, vector competence, the daily survival rate of a vector, the extrinsic incubation period, and the probability that the vectors survive this period [25,27,28]. Vectorial capacity is thus specific to a given vector population at prevailing climatic conditions in a specific area at a specific time [12].

To our knowledge, our study is the first to investigate whether *An. plumbeus* mosquitoes are competent for JEV. Vector competence studies were conducted at two different temperatures during incubation, namely a constant 25 °C condition next to a 25/15 °C day/night temperature gradient. The temperature gradient was chosen to verify whether such temperature condition, representative of northwestern European summers [29], leads to significant differences in infection, dissemination and/or transmission rates. Since pigs are important amplification hosts, we furthermore studied whether *An. plumbeus* would readily feed on pigs. Thus, if *An. plumbeus* is competent for JEV and would feed on pigs, they could contribute significantly to the spread of this virus upon introduction in our region.

## 2. Materials and Methods

### 2.1. Mosquito Collection

Larvae (for vector competence studies) and adult *An. plumbeus* mosquitoes (for the assessment of blood feeding on pigs) were collected at De Vortebossen, Ruiselede, Belgium (51°04′15.0″ N 3°22′00.1″ E). The field-collection was carried out in 2021 and 2022. Larvae were collected in plastic containers with water from the collection site and were fed with brewer’s yeast tablets and fish food. Larvae and adults were kept at 24 °C, 70% relative humidity with a 16:8 h light–dark cycle, and a 10% sucrose solution was provided to the adults as a food source.

### 2.2. Virus Production and Titration

For our experiments, a pre-existing eighth passage of JEV genotype 3 Nakayama strain with a titer of 10^7.7^ TCID50/mL was used for our experiments. For the detailed protocol, we refer to a previous study by our team [30].

### 2.3. Oral Infection and Incubation of Mosquitoes

Oral infection, dissection and salivation were performed according to the standardized conditions described in [12] and applied in [30]. In short, three- to seven-day-old F0 generation *An. plumbeus* mosquitoes were deprived from any food source for 24 h. Subsequently, mosquitoes were transported to our BSL-3 facilities. Blood feeding was performed for one hour using the Hemotek system with pig intestines (Butcher Burms, Ghent and Butcher Andy, Pittem, Belgium) as a membrane. The blood meal consisted of a JEV suspension (genotype 3 Nakayama strain) with chicken blood (Van-O-Bel poultry slaughterhouse + 5 μM ATP) at a 1:2 ratio. The viral titer in the blood meal was 10^6.39^ TCID50/mL at T0 and remained >10^5.5^ TCID50/mL at T1. After blood feeding, blood-fed females were selected, transferred to a bugdorm mosquito cage and kept at either a constant 25 °C or a 25/15 °C day/night temperature gradient (with gradual increases/decreases in temperature and light intensity for 4 h) with 70% relative humidity and a 16:8 h light–dark cycle. A 10% sucrose solution was provided during the 14-day incubation period.

### 2.4. Mosquito Salivation and Dissection

After 14 days (14 dpi), the surviving JEV-exposed mosquitoes were cold anesthetized for approximately two minutes, and samples were collected to determine infection, dissemination and transmission rate according to the protocol described previously [30]. Briefly, the mosquito body was collected for infection; the legs, wings and head were collected for dissemination, and saliva was collected for transmission. For saliva collection, *An. plumbeus* mosquitoes (without legs and wings) were fixed to a glass slide with double-sided adhesive tape, and their proboscis was placed in a 10 µL pipette tip filled with 5 μL of FBS (Merck Life Science, Hoeilaart, Belgium) containing 10% sugar which was held in place using modeling clay (Figure 1). These glass slides were placed on an angled plate (±30°) to allow the mosquitoes to salivate for 30 min.

A positive control for salivation was applied which involved observing the movements of the maxillary palps and stylets during salivation, the formation of bubbles in the media [31,32] and whether the abdomen of the mosquito was enlarged indicating they ingested the FBS and thus have salivated (a method kindly recommended by dr. Lanjiao Wang of the KU Leuven). After 30 min, the content of the tip was transferred into an Eppendorf containing 45 μL DMEM with 2% antibiotics (Antibiotic Antimycotic Solution 100×, Sigma-Aldrich, St. Louis, MO, USA) and 2% FBS.

### 2.5. JEV Detection

#### 2.5.1. RT-qPCR Analysis

Stainless steel beads were added to the mosquito bodies and to the head-wings-legs in DMEM with antibiotics, and homogenization was conducted using a Tissuelyser (Tissuelyser II, Qiagen, Hilden, Germany). RNA was extracted using the QIAamp viral RNA mini kit (Qiagen) according to the manufacturer’s protocol. Saliva samples were adjusted to 140 µL (25 µL sample + 115 µL nuclease free water), as the procedure is optimized for use with 140 µL samples. RNA was used directly for RT-qPCR analysis or stored at −80 °C until use. The RT-qPCR mixture contained AgPath-ID™ One-Step RT-PCR Reagents, 800 nM of each of the JEV NS2A primers (forward: 5′-AGCTGGGCCTTCTGGT-3′ and reverse: 5′-CCCAAGCATCAGCACAAG-3′) and 400 nM of the probe (FAM-CTTCGCAAGAGGTGGACGGCCA-TAMRA) [33], and 5 μL of the RNA sample and positive and negative controls was added to each run. Samples were run on a LightCycler480 according to the following temperature program: 45 °C for 10 min and 95 °C for 10 min, followed by 45 cycles at 95 °C for 15 s and 60 °C for 45 s. CT-values ≤ 40 and with curves that showed an exponential amplification were considered positive. Each sample was tested for amplification of beta-actin, a housekeeping gene, as an internal control. A similar RT-qPCR mixture was prepared for detection of JEV. Beta-actin primers (forward: 5′-GTRTGGATYGGHGGCTCCATY-3′ and reverse: 5′-GACTCRTCRTACTCCTGCTTG-3′) were present at 480 nM and the probe (FAM-ACCTTCCAGCAGATGTGGATC) at 320 nM in the mix. As the CT-values for this housekeeping gene were highly stable between samples (mean CT ± SEM in infection and dissemination at 25 °C: 26.32 ± 0.2661 and 27.54 ± 0.2980, respectively; mean CT ± SEM in infection and dissemination at 25/15 °C: 25.39 ± 0.2094 and 27.63 ± 0.01000, respectively), we reported the observed CT-values for JEV in the Section 3.

#### 2.5.2. Virus Isolation

RT-qPCR-positive samples were additionally subjected to virus isolation attempts on Vero cells. We refer the reader to our previous study for the detailed protocol [30].

### 2.6. Anopheles plumbeus Blood Feeding on Pigs

To determine whether *An. plumbeus* would feed on pigs, female mosquitoes, collected in the field and deprived of any food source for 24 h, were given the opportunity to feed on an anesthetized pig at our experimental center in Machelen. To this end, the mosquitoes were divided into four cages covered with mesh allowing them to bite through. The cages were placed on the pig in the abdominal region or on the ears and were kept in place for 1 h using tape. Afterwards, the mosquitoes were placed in the freezer, and blood-fed mosquitoes were counted by visual inspection.

### 2.7. Statistical Analysis

Fisher’s exact tests were used to determine whether infection, dissemination and transmission rates differed significantly between the two temperature conditions. Unpaired *t*-tests were used to determine whether significant differences in the CT-values occurred between the two temperature conditions and between the CT-values in infection from dissemination positive and dissemination negative mosquitoes in the 25 °C condition. Linear regression was implemented to see whether a correlation existed between the CT-values in infection and dissemination at the 25 °C condition. Statistical analyses were done using GraphPad Prism 9. *p*-values < 0.05 were considered to be significant.

## 3. Results

### 3.1. Infection, Dissemination and Transmission Rates

#### 3.1.1. Incubation at a Constant 25 °C Temperature

Of the approximately 1000 mosquitoes that were allowed to feed on an infected blood meal, 293 *An. plumbeus* fed (29.3%). These were subsequently incubated at 25 °C. Ninety-one (30.7%) *An. plumbeus* mosquitoes survived this 14 day incubation period. We found an infection rate of 34.1% (31/91), and of these 31 infected mosquitoes, JEV disseminated in 21, resulting in a dissemination rate of 67.7%. Of the dissemination-positive mosquitoes, three were found positive by RT-qPCR in saliva which results in a transmission rate of 14.3% (3/21). However, only 12 of the mosquitoes with a disseminated infection showed visible evidence of salivation. If we only consider these mosquitoes with visual proof of salivation, the transmission rate amounts to 25%. The overall transmission efficiency (i.e., positive saliva samples upon total mosquitoes blood-fed) was 3.3% for *An. plumbeus* (Table 1; Figure 2) when incubated at 25 °C.

#### 3.1.2. Incubation at a 25/15 °C Day/Night Temperature Gradient

In a second experiment, an infected blood meal was presented to approximately 900 *An. plumbeus* and 158 (17.6%) mosquitoes fed. These were subsequently incubated at a 25/15 °C temperature gradient. On completion of the 14-day incubation period, 47 (29.7%) *An. plumbeus* mosquitoes survived (Table 1). An infection rate of 25.5% was detected, which did not differ significantly from the infection rate at the constant 25 °C condition (*p*-value: 0.3362). The dissemination rate however was significantly lower at the temperature gradient, i.e., 16.7% compared to 67.7% in the constant temperature condition (*p*-value: 0.006). JEV RNA was not detected in the saliva of either of the two dissemination-positive mosquitoes, resulting in a transmission rate and transmission efficiency of 0% (Figure 2).

### 3.2. Viral Loads Found in Infection, Dissemination and Transmission

Viral loads, based on CT-values, varied greatly between individual mosquitoes in both infection and dissemination. No significant difference was found between the mean CT of 30.41 (22.24 to 40) in infection at the constant temperature and CT 31.36 (23.57 to 39.5) at the temperature gradient (*p*-value: 0.7872) (Figure 3A). This was also the case for dissemination, with a mean CT of 33.19 (24.04 to 38.37) at the constant temperature and CT 36.74 (35.61 to 37.87) at the gradient temperature condition (*p*-value: 0.2782) (Figure 3B).

When incubated at a constant 25 °C, a significant difference was detected between mean CT in infection in mosquitoes with a disseminated infection (28.58) and without a disseminated infection (34.25) (*p*-value: 0.0093), indicating that a higher viral load in infection may predict whether JEV will disseminate. When performing a linear regression analysis between the CT-values in infection and dissemination of the individual mosquitoes at the constant 25 °C temperature condition (Figure 4), we saw that the slope of the regression was significantly different from zero (*p*-value: 0.0091), indicating that there is a correlation between the viral load in the midgut and the rest of the mosquitoes’ body (head, wings and legs) at day 14 post-infection.

It can also be noticed that the three mosquitoes with JEV-RNA in their saliva (green dots) have low CT-values both in infection and dissemination (located at the bottom left of the correlation graph), which would thus indicate virus load-dependent passage of the salivary gland escape barrier.

All JEV samples that were positive in RT-qPCR were assayed in a virus isolation test, and the majority (72.5%) were confirmed to contain infectious JEV (Table 2). Virus isolation failed to confirm the three RT-qPCR positive saliva samples, although this is not unexpected given the high CT-values in transmission (CT 39.38 to 40), indicating a low amount of virus in the saliva (Figure 3C).

### 3.3. Anopheles plumbeus Blood Feeding on Pigs

Small cages with female *An. plumbeus* collected in the field were attached to an anesthetized pig. After one hour, the mosquitoes were cold-euthanized, and their abdomens were inspected visually to determine how many had taken a blood meal from the sedated pig. Of the 102 female *An. plumbeus*, 33 (32%) were engorged. This proves that *An. plumbeus* readily feeds on pigs when the opportunity occurs.

## 4. Discussion

*Anopheles plumbeus* is increasingly being detected in Europe [34]. Their widespread occurrence in addition to their spread to more urban environments by adapting to artificial breeding sites enables them to increasingly interact with humans and livestock. This makes them important as possible vectors in the transmission of vector-borne zoonotic diseases. In the past, they have already been implicated in the field and/or laboratory transmission of malaria and West Nile virus [7]. Our study is the first to show that *An. plumbeus* is a competent vector for JEV which increases the list of potential vectors for JEV to 11: *Aedes detritus*, *Aedes dorsalis*, *Aedes japonicus*, *Aedes kochi*, *Aedes nigromaculis*, *Aedes notoscriptus*, *Culiseta annulata*, *Culiseta incidens*, *Culiseta inornate*, *Verrallina funereal* and now thus *An. plumbeus* [12]. These potential vectors are all species with a proven transmission in the laboratory though without having been found JEV-positive in the field. Once virus is detected in mosquitoes of a certain species in the field, it qualifies as a known vector species. Currently 17 species are known vectors for JEV: *Aedes albopictus*, *Aedes vexans*, *Aedes vigilax*, *Anopheles tessellatus*, *Armigeres subalbatus*, *Culex annulirostris*, *Culex bitaeniorhynchus*, *Culex fuscocephala*, *Culex gelidus*, *Culex pipiens*, *Culex pipiens pallens*, *Culex pseudovishnui*, *Culex quinquefasciatus*, *Culex sitiens*, *Culex tarsalis*, *Culex tritaeniorhynchus* and *Culex vishnui.*

The observed transmission efficiency of *An. plumbeus* for JEV was 3.3% (3/91) at the 25 °C temperature condition. This result was obtained using highly relevant F0 mosquitoes which emerged from field-collected larvae. A similar transmission efficiency was recently reported by our research group for JEV in F0 *Culex pipiens* mosquitoes [30]. Other vector competence studies with *Culex pipiens* for JEV [35,36] had resulted in noticeably higher transmission efficiencies. However, these studies used colonized mosquitoes which are adapted to laboratory conditions and genetically and phenotypically less diverse, which might lead to an overestimation of the transmission efficiency.

To assess the potential vector role of *An. plumbeus* under current climatic conditions in northwestern Europe, we applied a day/night temperature gradient of 25/15 °C, simulating Belgian summer conditions. Infection, dissemination and transmission rates obtained at this gradient condition were compared with the rates at the constant 25 °C temperature condition. This revealed that the vector competence of *An. plumbeus* is temperature dependent, as we found a significantly lower dissemination rate (16.7 versus 67.7%; *p*-value: 0.006) and no transmission at the temperature gradient condition. Consequently, our results suggest that it is unlikely that *An. plumbeus* would transmit JEV upon an accidental introduction into northwestern Europe under current climate conditions. This might change if climate changes causes higher average temperatures and/or the occurrence of periods with exceptionally high temperatures.

Regarding viral load, the correlation between CT-values in infection and dissemination at the constant 25 °C condition (*p*-value of the slope of the regression 0.0091) indicates that a higher viral load in infection is more likely to lead to a higher viral load in dissemination. In addition, we found a significant difference between the mean CT-values in infection of mosquitoes with and without a disseminated infection (*p*-value: 0.0093), suggesting that a higher viral load in the midgut increases the chance that JEV will succeed to cross the midgut escape barrier and lead to a disseminated infection. Nevertheless, it should be noted that the CT-values in infection vary greatly between the individual mosquitoes in which the virus disseminated and that also mosquitoes with a disseminated infection were observed amongst those with very low viral loads in infection. Additional research would be needed to determine the factors influencing the crossing of the midgut escape barrier. On the other hand, our study demonstrated that a higher viral load in dissemination increases the likelihood of transmission of the virus to the saliva. The three mosquitoes in which transmission was demonstrated had a high viral load in secondary tissues, suggesting a viral load dependent passing of the salivary gland escape barrier.

Another observation in our experiments is the low survival of the mosquitoes after infection at both temperatures (30.7 and 29.7%). Part of the blood-fed mosquitoes died during the incubation period, although they were maximum 20 days old (3- to 7-day-old mosquitoes + <14 days incubation), which contrasts with the reported lifespan of *An. plumbeus* of up to 2 months [3]. Mortality in this study was also remarkably higher than when working with *Culex pipiens* using the same protocol, in a previous study [30]. A possible explanation could be that *An. plumbeus* has difficulties adapting to artificial environments. Knowing that they originate from forested areas, they may not thrive as well in a laboratory environment as urban Culicid species, such as *Culex pipiens*, the common house mosquito.

An important aspect of vectorial capacity that we investigated was whether *An. plumbeus* would feed on pigs, the main amplification host of JEV. We demonstrated that 33 of the 102 field-collected female mosquitoes fed on a pig when given the opportunity. We thus showed for the first time that pigs can be added to the already wide host range of *An. plumbeus*, which already consisted of birds, horses, cattle and humans. We did not investigate *An. plumbeus*’ actual host species preference by, for example, providing the choice of feeding on birds, pigs or cattle. However, host preference is influenced by numerous factors. Some of these have a genetic basis, although the presence and density of a host species plays an important role as well. If a certain host species is abundantly available, a specific mosquito species will not unnecessarily search further [37].

## 5. Conclusions

Our study was the first to demonstrate that Belgian *An. plumbeus* mosquitoes are competent vectors for JEV and will blood feed on pigs, the main amplifying host of JEV. Combined with its known host range including birds, horses and cattle and its aggressive behavior towards humans, this suggests they could be a high-capacity vector for JEV in our region. However, under current temperature conditions, their role as a vector for JEV is unlikely as we obtained a low dissemination rate and no transmission when infected mosquitoes were incubated at a temperature gradient reflecting current summer conditions in Belgium. This could however change if temperatures were to rise due to climate change, as we did find JEV in saliva of mosquitoes incubated at a constant 25 °C.

## Figures and Tables

**Figure 1 microorganisms-11-01386-f001:**
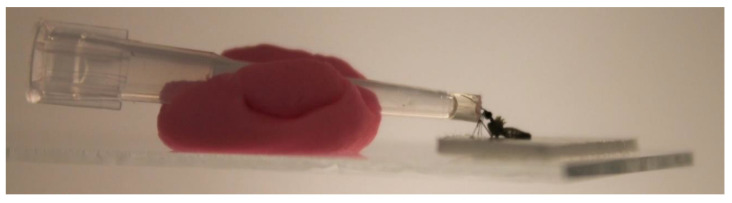
Mosquito salivation set-up.

**Figure 2 microorganisms-11-01386-f002:**
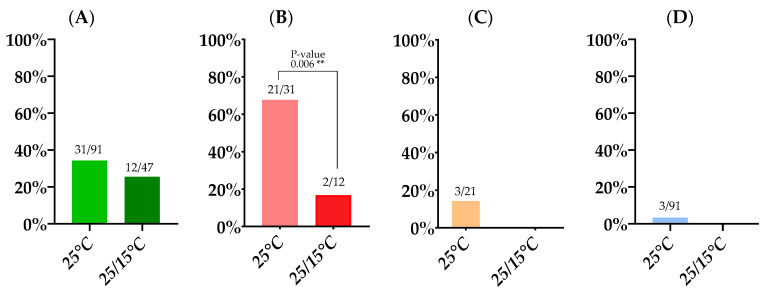
Infection rate (**A**), dissemination rate (**B**), transmission rate (**C**), and transmission efficiency (**D**) at the two temperature conditions. Absolute numbers are displayed above bars. Fisher’s exact tests were performed to check differences between rates at both temperature conditions. ** indicates a *p*-value ≤ 0.01.

**Figure 3 microorganisms-11-01386-f003:**
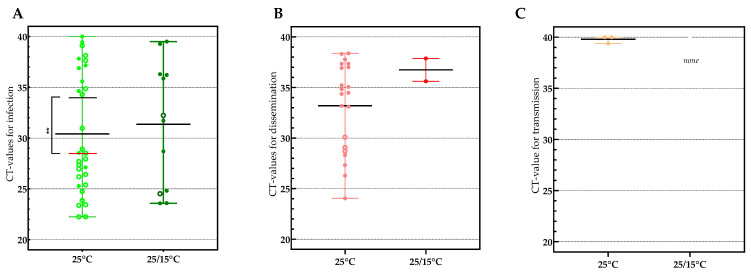
Viral loads (CT-values) found in samples of *An. plumbeus* mosquitoes found positive for infection (**A**), dissemination (**B**) and transmission (**C**). Mean CT’s are indicated by horizontal black lines. Open circles in panel (**A**) indicate individual mosquitoes that tested positive in dissemination and in (**B**) mosquitoes that tested positive in transmission. Filled dots in panel (**A**) indicate individual mosquitoes that tested negative in dissemination and in (**B**) in transmission. An unpaired *t*-test was used to determine whether a significant difference occurred between the CT-values in infection from dissemination positive and dissemination negative mosquitoes in the 25 °C condition. ** indicates a *p*-value ≤ 0.01.

**Figure 4 microorganisms-11-01386-f004:**
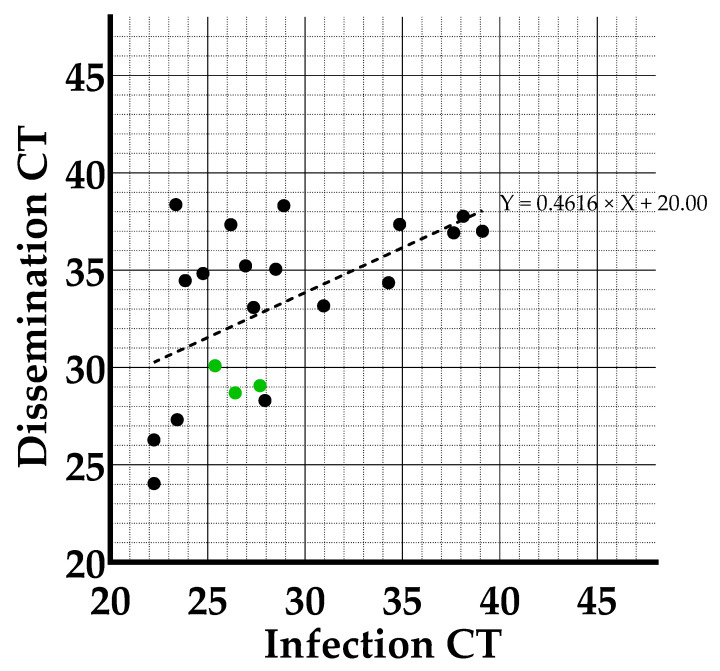
Correlation between CT-values for infection and dissemination in *An. plumbeus* at the constant 25 °C temperature condition implemented during incubation. Green dots indicate mosquitoes which tested positive in transmission.

**Table 1 microorganisms-11-01386-t001:** Vector competence of *An. plumbeus* for JEV at two temperature conditions. Results are based on RT-qPCR analyses. Rates are mentioned in percentages, and absolute numbers are mentioned between brackets.

	14 Day Survival Rate	Infection Rate	Dissemination Rate	Transmission Rate	Transmission Efficiency
25 °C	30.7% (91/293)	34.1% (31/91)	67.7% (21/31)	14.3% (3/21)	3.3% (3/91)
25/15 °C	29.7% (47/158)	25.5% (12/47)	16.7% (2/12)	0% (0/2)	0% (0/47)

**Table 2 microorganisms-11-01386-t002:** Virus isolation of samples positive in RT-qPCR for infection, dissemination and transmission. All ratios are given in percentages with the respective absolute numbers between brackets.

	Infection RT-qPCR Positive Samples Confirmed by Isolation	Dissemination RT-qPCR Positive Samples Confirmed by Isolation	Transmission RT-qPCR Positive Samples Confirmed by Isolation
25 °C	96.7% (30/31)	33.3% (7/21)	0% (0/3)
25/15 °C	91.7% (11/12)	100% (2/2)	No RT-qPCR positives

## Data Availability

All available data has been described within the manuscript.

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
