# Peer review of "Belgian Anopheles plumbeus Mosquitoes Are Competent for Japanese Encephalitis Virus and Readily Feed on Pigs, Suggesting a High Vectorial Capacity"

_microorganisms, 2023, doi:10.3390/microorganisms11061386_

Round 1
Reviewer 1 Report
The study entitled “Belgian Anopheles plumbeus mosquitoes are competent for Japanese encephalitis virus and readily feed on pigs, suggesting a high vectorial capacity” authored by Claudia Van den Eynde, Charlotte Sohier, Severine Matthijs and Nick De Regge, is devoted to highlight the Anopheles plumbeus mosquitoes as competent vectors for JEV, blood feed on pigs, the main amplifying host of JEV. The presented research is of scientific importance and contains the potential for applied use. My main complaints are following:
1) Plotting directly ct value has become an old and outdated way. I want to know why didn’t use any reference gene or absolute quantification in qPCR? Then either ddct (relative expression) or 2ddct/2^ddct (fold change) calculated value which will come that should be plotted for the representation.
2) You had constantly used Anopheles plumbeus throughout the manuscript. You don’t need to repeat the full name each time instead can use “An. plumbeus”.
Author Response
We thank the reviewer for the constructive criticism. We have addressed all comments and made modifications to the manuscript as documented below and with track changes in the revised manuscript. During the revision, we observed a calculation error in the mean CT-value for infection of the mosquitoes without a disseminated infection, which was adjusted accordingly. If there are any suggestions for the revised version, please do not hesitate to contact us. We appreciate the thorough analysis and hope that this improved version will be published later in Microorganisms.
- Plotting directly ct value has become an old and outdated way. I want to know why didn’t use any reference gene or absolute quantification in qPCR? Then either ddct (relative expression) or 2ddct/2^ddct (fold change) calculated value which will come that should be plotted for the representation.
The analyses we conducted did involve the use of an internal control (beta-actin). However, since the CT-values in infection and dissemination for this internal reference gene were highly similar in all tested samples (means with standard errors mentioned below), we chose to use the observed CT-values as such and not ddCt values.
The following was added to the M&M (line 159-166): Each sample was tested for amplification of beta-actin, a housekeeping gene, as an internal control. A similar RT-qPCR mixture was prepared as for detection of JEV. Beta-actin primers (forward: 5’-GTRTGGATYGGHGGCTCCATY-3’ and reverse: 5’-GACTCRTCRTACTCCTGCTTG-3’) were present at 480 nM and the probe (FAM-ACCTTCCAGCAGATGTGGATC) at 320 nM in the mix. As the CT-values for this housekeeping gene were highly stable between samples (mean CT+/-SEM in infection and dissemination at 25°C: 26.32+/- 0.2661 and 27.54+/-0.2980, respectively; mean CT+/-SEM in infection and dissemination at 25/15°C: 25.39+/- 0.2094 and 27.63+/-0.01000, respectively), we reported the observed CT-values for JEV in the results section.
Results for beta-actin :
- 25°C condition:
- mean CT in infection: 26.32, SEM: 0.2661
- mean CT in dissemination: 27.54, SEM: 0.2980
- 25/15°C condition:
- mean CT in infection: 25.39, SEM: 0.2094
- mean CT in dissemination: 27.63, SEM: 0.01000
- You had constantly used Anopheles plumbeus throughout the manuscript. You don’t need to repeat the full name each time instead can use “ plumbeus”.
Implemented. Anopheles plumbeus was changed to An. plumbeus throughout the manuscript.
Reviewer 2 Report
This is a well-written, well-designed experiment to show if Anopheles plumbeus can, or are likely to serve, as vector or JEV. The only section that I feel can be improved is materials and methods section 2.4.
Specifically: How long does the anesthesia last in order to get the insect manipulated and in place for the salivation test? Once in place, how long does the mosquito have to salivated and ingest the sugar-containing FBS? Was this a standardized time?
L128-130 is confusing. Perhaps change to "...FBS containing 10% sugar which was held in place using modeling clay." As written now, it suggests that the proboscis is held in place with the clay and based on the photo, that is not the case.
How was the saliva collected and how was it tested for JEV? This information is not clearly stated in the methods section.
Author Response
We thank the reviewer for the constructive criticism. We have addressed all comments and made modifications to the manuscript as documented below and with track changes in the revised manuscript. During the revision, we observed a calculation error in the mean CT-value for infection of the mosquitoes without a disseminated infection, which was adjusted accordingly. If there are any suggestions for the revised version, please do not hesitate to contact us. We appreciate the thorough analysis and hope that this improved version will be published later in Microorganisms.
- Specifically: How long does the anesthesia last in order to get the insect manipulated and in place for the salivation test? Once in place, how long does the mosquito have to salivated and ingest the sugar-containing FBS? Was this a standardized time?
Implemented. The following was added to the materials and methods section (line 124-125): After 14 days (14 dpi), the surviving JEV-exposed mosquitoes were cold anesthetized for approximately two minutes,
And (line 132-133): These glass slides were placed on an angled plate (± 30°) to allow the mosquitoes to salivate for 30 minutes.
- L128-130 is confusing. Perhaps change to "...FBS containing 10% sugar which was held in place using modeling clay." As written now, it suggests that the proboscis is held in place with the clay and based on the photo, that is not the case.
Implemented.
- How was the saliva collected and how was it tested for JEV? This information is not clearly stated in the methods section.
Implemented, the following was added to clarify how saliva was collected (line 141-143): After 30 minutes, the content of the tip was transferred into an Eppendorf containing 45 μl DMEM with 2% antibiotics (Antibiotic Antimycotic Solution 100×, Sigma-Aldrich) and 2% FBS.
In addition, this was included to clarify how saliva was tested for JEV (line 149-151): Saliva samples were adjusted to 140 µl (25 µl sample + 115 µl nuclease free water), as the procedure is optimized for use with 140 µl samples.